# Balrog: A universal protein model for prokaryotic gene prediction

**Markus J. Sommer**[1,2]*, **Steven L. Salzberg**[1,2,3]

**1** Department of Biomedical Engineering, Johns Hopkins University, Baltimore, Maryland, United States of America, **2** Center for Computational Biology, Johns Hopkins University, Baltimore, Maryland, United States of America, **3** Departments of Computer Science and Biostatistics, Johns Hopkins University, Baltimore, Maryland, United States of America

* markusjsommer@gmail.com

## Abstract

Low-cost, high-throughput sequencing has led to an enormous increase in the number of sequenced microbial genomes, with well over 100,000 genomes in public archives today. Automatic genome annotation tools are integral to understanding these organisms, yet older gene finding methods must be retrained on each new genome. We have developed a universal model of prokaryotic genes by fitting a temporal convolutional network to amino-acid sequences from a large, diverse set of microbial genomes. We incorporated the new model into a gene finding system, Balrog (Bacterial Annotation by Learned Representation Of Genes), which does not require genome-specific training and which matches or outperforms other state-of-the-art gene finding tools. Balrog is freely available under the MIT license at https://github.com/salzberg-lab/Balrog.

**Data Availability Statement:** All genome files are available from the NCBI GenBank database (accession numbers are provided in S1–S3 Appendices)

**Funding:** This work was supported in part by grants awarded to SLS: R35–GM130151 and R01-HG006677 from the National Institutes of Health

## Author summary

Annotating the protein-coding genes in a newly sequenced prokaryotic genome is a critical part of describing their biological function. Relative to eukaryotic genomes, prokaryotic genomes are small and structurally simple, with 90% of their DNA typically devoted to protein-coding genes. Current computational gene finding tools are therefore able to achieve close to 99% sensitivity to known genes using species-specific gene models. Though highly sensitive at finding known genes, all current prokaryotic gene finders also predict large numbers of additional genes, which are labelled as "hypothetical protein" in GenBank and other annotation databases. Many hypothetical gene predictions likely represent true protein-coding sequence, but it is not known how many of them represent false positives. Additionally, all current gene finding tools must be trained specifically for each genome as a preliminary step in order to achieve high sensitivity. This requirement limits their ability to detect genes in fragmented sequences commonly seen in metagenomic samples. We took a data-driven approach to prokaryotic gene finding, relying on the large and diverse collection of already-sequenced genomes. By training a single, universal model of bacterial genes on protein sequences from many different species, we were able to match the sensitivity of current gene finders while reducing the overall number of

https://www.nih.gov/. The funders had no role in study design, data collection and analysis, decision to publish, or preparation of the manuscript.

**Competing interests:** The authors have declared that no competing interests exist.

gene predictions. Our model does not need to be refit on any new genome. Balrog (Bacterial Annotation by Learned Representation of Genes) represents a fundamentally different yet effective method for prokaryotic gene finding.

## Introduction

One of the most important steps after sequencing and assembling a microbial genome is the annotation of its protein-coding genes. Methods for finding protein-coding genes within a prokaryotic genome are highly sensitive, and thus have seen little change over the past decade. Widely used prokaryotic gene finders include various iterations of Glimmer [1, 2], GeneMark [3, 4], and Prodigal [5], all of which are based on Markov models and which utilize an array of biologically-inspired heuristics. Each of these previous methods requires a bootstrapping step to train its internal gene model on each new genome. This requirement also limits their ability to detect genes in fragmented sequences commonly seen in metagenomic samples [6].

The lack of recent advances in *ab initio* bacterial gene finding tool development is partly due to the perception that bacterial gene finding is a solved problem. Currently available tools achieve near 99% sensitivity for known genes (i.e., genes with a functional annotation), so there appears to be little room for improvement. However, all current software tools predict hundreds or thousands of "extra" genes per genome, i.e., genes that do not match any gene with a known function and are usually given the name "hypothetical protein." Many of these hypothetical genes likely represent genuine protein coding sequences, but many others may be false positive predictions. It is difficult if not impossible to prove that a predicted open reading frame is not a gene; thus these hypothetical proteins have remained in genome annotation databases for many years. However, systematically annotating false positives as genes may create problems for downstream analyses of genome function [7].

In line with evaluation metrics used by other gene finders, if a program can find nearly all true positive genes while predicting fewer genes overall, it is reasonable to assume this is primarily due to a reduction in false positive predictions [2, 5]. Thus, we would prefer a method that makes fewer overall predictions while retaining very high sensitivity to known genes.

Currently available gene finders were developed in the late 1990's and 2000's, when relatively few prokaryotic genomes were available. Today, tens of thousands of diverse bacterial genomes from across the prokaryotic tree of life have been sequenced and annotated. We hypothesized that it should therefore be feasible to build a data-driven gene finder by training a machine learning model on a large, diverse collection of high-quality prokaryotic genomes. The program could then be applied, without any further re-training or adjustment, to find genes in any prokaryotic species. Balrog was developed with this strategy in mind. In the experiments below, we show that Balrog, when trained on all high-quality prokaryotic genomes available today, matches the sensitivity of current state-of-the-art gene finders while reducing the total number of hypothetical gene predictions. By integrating protein-coding gene predictions from Balrog, standard prokaryotic annotation and analysis pipelines such as NCBI PGAP (Prokaryotic Genome Annotation Pipeline) [8], MGnify [9], or Prokka [10] may improve their genome annotation quality.

## Results

### Gene prediction sensitivity

We compared the performance of Balrog, Prodigal, and Glimmer3 by running each tool with default settings on a test set of 30 bacteria and 5 archaea that were not included in the Balrog

training set. Following the conventions established in multiple previous studies, we considered a protein-coding gene to be known if it was annotated with a name not including "hypothetical" or "putative." In standard annotation pipelines, proteins are labeled hypothetical if they have no significant match to known protein sequences and are not otherwise covered by a standard naming rule [11]. For most bacterial genomes, more than two-thirds of their annotated genes fall into the "known" category, with the rest being hypothetical. The hypothetical genes include a mixture of true genes and false positive predictions. In our experiments, we measured the total number of genes predicted in each genome and calculated the sensitivity of each program to known, non-hypothetical genes. Predictions were considered correct if the stop codon was correctly predicted, i.e., if the 3' position of the gene was correct. Results for this gene finder comparison can be found in Table 1.

All three tools achieved similar sensitivity on the bacterial genomes in the test set. On average, Balrog found 2 non-hypothetical genes fewer than Prodigal (2,248 vs. 2,250) and 3 genes more than Glimmer3 (2,248 vs. 2,245). This represents a difference of less than 0.1% in sensitivity. Balrog predicted the fewest genes overall, reducing the number of "extra" gene predictions by 11% vs. Prodigal (664 vs. 747) and 30% vs Glimmer3 (664 vs. 949).

Balrog predicted more genes than Prodigal for only one bacterial genome, *E. coli* K-12 MG1655 (the standard laboratory strain). On that genome, Balrog predicted 3 more extra genes than Prodigal, but at the same time it found 43 more true annotated genes. It is worth noting here that all organisms in the Escherichia and Shigella genuses were excluded from the Balrog training data set.

On the five genomes in the archaea test set, we observed more pronounced differences in the number of extra gene predictions. Glimmer3 found the most known genes, averaging 1670, versus 1663 for Prodigal and 1661 for Balrog. However, Balrog predicted the fewest genes overall, 18% fewer extra genes than Prodigal and 40% fewer than Glimmer3.

Similar results were observed when the gene model was trained on a set excluding organisms sharing a family, rather than a genus, with any organism in the test set. On average, the gene model achieved sensitivity of 98.12% with family excluded vs. 98.15% with genus excluded (2247 vs. 2248 genes) in bacteria and 97.50% vs. 97.44% (1662 vs. 1661) in archaea. The family-excluded model predicted on average 25 more extra genes than the genus-excluded model in bacteria (689 vs. 664) and 32 more in archaea (597 vs. 565).

## Materials and methods

### Training and testing data

In selecting genomes on which to train our gene model, we aimed to cover as much microbial diversity as possible while limiting sequence redundancy. As a whole, currently available prokaryotic genomes are biased toward clinically relevant organisms. Many low-abundance environmental species may be absent from public databases, whereas organisms important to human disease may have full genomes for hundreds of closely related strains [12]. We cannot fully account for the missing diversity within available sequence databases (indeed, millions of bacterial species probably remain unsequenced), but to limit the impact of highly-overrepresented species, we randomly selected only one genome for each bacterial and archaeal species within the Genome Taxonomy Database (GTDB, https://gtdb.ecogenomic.org) for the training set [13]. Only high-quality genomes were selected, defined by GTDB as over 90% complete with less than 5% contamination. Because high-quality protein annotations were also necessary, we required selected genomes to be available in RefSeq or GenBank with the tag "Complete Genome" and without the tag "Anomalous assembly."

**Table 1. Non-hypothetical gene prediction comparison.**

| Genome | | | Balrog | | | Prodigal | | | Glimmer3 | | |
|---|---|---|---|---|---|---|---|---|---|---|---|
| Bacteria | GC | genes | 3′ matches | | extra | 3′ matches | | extra | 3′ matches | | extra |
| | % | # | # | % | # | # | % | # | # | % | # |
| T. narugense | 30 | 1570 | **1559** | 99.3 | **271** | 1557 | 99.2 | 302 | **1559** | 99.3 | 367 |
| C. fetus | 31 | 1486 | **1476** | 99.3 | **216** | 1475 | 99.3 | 248 | 1473 | 99.1 | 279 |
| T. wiegelii | 33 | 2359 | 2265 | 96.0 | **505** | 2255 | 95.6 | 557 | **2267** | 96.1 | 715 |
| Nat. thermophilus | 34 | 2419 | 2397 | 99.1 | **479** | 2401 | 99.3 | 554 | **2403** | 99.3 | 648 |
| D. thermolithotrophum | 34 | 1360 | **1336** | 98.2 | **197** | 1336 | 98.2 | 220 | 1332 | 97.9 | 257 |
| D. thermophilum | 37 | 1630 | 1607 | 98.6 | **250** | 1609 | 98.7 | 281 | **1609** | 98.7 | 333 |
| P. UFO1 | 38 | 3873 | **3834** | 99.0 | **725** | 3829 | 98.9 | 970 | 3831 | 98.9 | 1134 |
| T. takaii | 40 | 1496 | 1484 | 99.2 | **322** | **1486** | 99.3 | 373 | 1485 | 99.3 | 422 |
| K. pacifica | 41 | 1608 | **1597** | 99.3 | **405** | 1596 | 99.3 | 441 | 1594 | 99.1 | 543 |
| B. bacteriovorus | 42 | 1897 | 1883 | 99.3 | **840** | **1887** | 99.5 | 921 | 1884 | 99.3 | 1027 |
| P. HL-130-GSB | 45 | 1882 | 1804 | 95.9 | **515** | 1809 | 96.1 | 604 | **1810** | 96.2 | 783 |
| C. thermautotrophica | 46 | 2137 | 2107 | 98.6 | **508** | **2116** | 99.0 | 595 | 2114 | 98.9 | 696 |
| A. aeolicus | 46 | 885 | **884** | 99.9 | **784** | 883 | 99.8 | 826 | 879 | 99.3 | 840 |
| M. thermoacetica | 49 | 2299 | 2227 | 96.9 | **679** | 2233 | 97.1 | 808 | **2238** | 97.3 | 1134 |
| Nov. thermophilus | 49 | 2850 | 2769 | 97.2 | **789** | 2754 | 96.6 | 929 | **2771** | 97.2 | 1103 |
| T. oceani | 49 | 1998 | 1941 | 97.1 | **305** | 1932 | 96.7 | 375 | **1943** | 97.2 | 533 |
| D. indicum | 50 | 2178 | 2152 | 98.8 | **461** | **2154** | 98.9 | 492 | 2134 | 98.0 | 679 |
| L. boryana | 50 | 4031 | 3947 | 97.9 | **1588** | **3956** | 98.1 | 1868 | 3953 | 98.1 | 2423 |
| D. multivorans | 51 | 3128 | 3061 | 97.9 | **667** | 3064 | 98.0 | 796 | **3065** | 98.0 | 1585 |
| E. coli K-12 MG1655 | 52 | 3529 | **3451** | 97.8 | 914 | 3408 | 96.6 | **911** | 3368 | 95.4 | 1110 |
| D. acetoxidans | 52 | 2322 | **2273** | 97.9 | **554** | 2268 | 97.7 | 698 | 2268 | 97.7 | 1165 |
| C. parvum | 54 | 1780 | **1753** | 98.5 | **301** | 1752 | 98.4 | 348 | 1746 | 98.1 | 489 |
| T. ammonificans | 56 | 1382 | 1373 | 99.3 | **306** | **1377** | 99.6 | 354 | 1373 | 99.3 | 362 |
| A. acidocaldarius | 58 | 2499 | 2393 | 95.8 | **617** | **2397** | 95.9 | 724 | **2397** | 95.9 | 908 |
| R. radiotolerans | 60 | 2196 | 2155 | 98.1 | **563** | **2166** | 98.6 | 608 | 2160 | 98.4 | 742 |
| D. desulfuricans | 62 | 2889 | 2849 | 98.6 | **578** | 2853 | 98.8 | 619 | **2854** | 98.8 | 858 |
| S. thermophilum | 63 | 2612 | 2564 | 98.2 | **652** | **2567** | 98.3 | 730 | 2562 | 98.1 | 847 |
| V. incomptus | 65 | 2498 | 2451 | 98.1 | **1131** | **2465** | 98.7 | 1176 | 2447 | 98.0 | 1540 |
| C. bipolaricaulis | 65 | 1022 | 997 | 97.6 | **237** | **1008** | 98.6 | 260 | 1000 | 97.8 | 286 |
| S. amylolyticus | 73 | 4880 | 4778 | 97.9 | **3631** | **4821** | 98.8 | 3887 | 4728 | 96.9 | 4789 |
| Averages: | 49 | 2289 | 2248 | 98.2 | **664** | **2250** | 98.3 | 747 | 2245 | 98.1 | 949 |
| Archaea | | | | | | | | | | | |
| M. ruminantium | 36 | 1710 | 1678 | 98.1 | **455** | 1682 | 98.4 | 517 | **1687** | 98.7 | 570 |
| A. GW2011 AR10 | 39 | 621 | 618 | 99.5 | **607** | **621** | 100.0 | 720 | **621** | 100.0 | 778 |
| M. sp. WWM596 | 46 | 2757 | 2567 | 93.1 | **840** | 2545 | 92.3 | 1123 | **2581** | 93.6 | 1999 |
| M. labreanum | 50 | 1390 | 1372 | 98.7 | **379** | 1370 | 98.6 | 446 | **1376** | 99.0 | 581 |
| H. lacusprofundi | 61 | 2047 | 2001 | 97.8 | **613** | **2017** | 98.5 | 731 | 2015 | 98.4 | 884 |
| Averages: | 46 | 1705 | 1661 | 97.4 | **565** | 1663 | 97.6 | 691 | **1670** | 97.9 | 949 |

"genes" refers to all protein-coding genes in the NCBI annotation where the description does not contain "hypothetical" or "putative." Genes with descriptions containing "hypoth" or "etical" are also excluded to catch the most common misspellings of hypothetical.

"3′ matches" counts the number of genes with stop sites exactly matching between the annotation and prediction on the same strand. "extra" counts the number of genes predicted by each program that do not share strand and stop site with an annotated non-hypothetical gene. The lowest number of extra genes and the highest number of 3′ matches are bolded for each organism.

From this set of high-quality complete genomes with gene annotations, 29 bacterial and five archaeal species were randomly selected to serve as a test set. *Escherichia coli* was also put in the test set because it is often used as a benchmark organism to compare gene finders. All genomes sharing a GTDB genus with any species in the test set were excluded from the training set. Full organism names and accession numbers for the testing data are available in S1 Appendix, while data for all organisms used to train the gene model are available in S2 Appendix. Though many gene sequences likely overlap between training and test data, we feel this test set should allow a reasonably conservative estimate of generalization error when predicting genes on a newly sequenced prokaryotic genome, which likely shares many gene sequences with previously seen genomes. Overall, this genome selection process yielded 3290 genomes in the training set and 36 in the test set. Additionally, a separate training set was constructed excluding all organisms sharing a family with any organism in the test set. This yielded 3085 genomes in the training set while the test set remained the same.

From all genomes, we extracted amino-acid sequences from annotated non-hypothetical genes. All genes with a description containing "hypothetical" or "putative" were removed from analysis, as many of these are not true genes but instead are the predictions of other gene finding programs. Additionally, genes with descriptions containing "hypoth" or "etical" were excluded in an effort to catch the most common misspellings of hypothetical. All non-hypothetical gene sequences were translated in all five alternative reading frames, and from these translations we extracted open reading frames (ORFs) longer than 100 amino acids to use as training examples of non-protein sequence.

We extracted amino-acid shingles (overlapping subsequences) in the 3' to 5' direction of length 100 and overlapping by 50 from all protein and non-protein sequences. These were used as positive and negative gene examples, respectively. In total, ≈27 gigabases (9 billion amino acids) of translated gene and non-gene sequence was generated to train the gene model.

## Training the gene model

A temporal convolutional network (TCN) was trained using the methods and open source Python framework of Bai et al. [14], slightly modified to enable binary classification of protein sequence. We use the state of the last node of the linear output layer as representative of the binary classifier, with a value close to 1 predicting a protein-coding gene sequence and 0 predicting an out-of-frame sequence. During training, binary cross-entropy loss was calculated on the state of this last node. Backpropagation from this loss minimizes gene prediction error based on the full context of our 100 amino-acid sequence shingles. This works because we set parameters such that the receptive field size of the network was sufficient to cover the whole length of a sequence shingle. Fig 1 shows an example TCN with each parameter explained.

During inference, we use the output from the pre-trained TCN to predict a single score for an ORF of any given length. To predict a single probability between 0 and 1, we combine all output scores from the TCN according to Eq 1, where $L$ is the length of the ORF and $p_i$ is the predicted gene probability by the TCN model at position $i$. This represents taking a weighted average predicted gene probability, then applying the logistic sigmoid function to map back from $(-\infty, \infty)$ to $(0, 1)$. This method has the effect of more heavily weighing TCN predictions that are close to 0 or 1 [15, 16]. We expect certain regions of a gene may contain recognizable protein sequence motifs, causing the TCN to predict a probability near 1. Other regions of a gene may contain little recognizable information, causing the TCN to predict near 0.5. By combining scores using this function, a single prediction near 1, caused by a recognizable protein motif, can force the combined gene score closer to 1. Simply put, this equation allows us

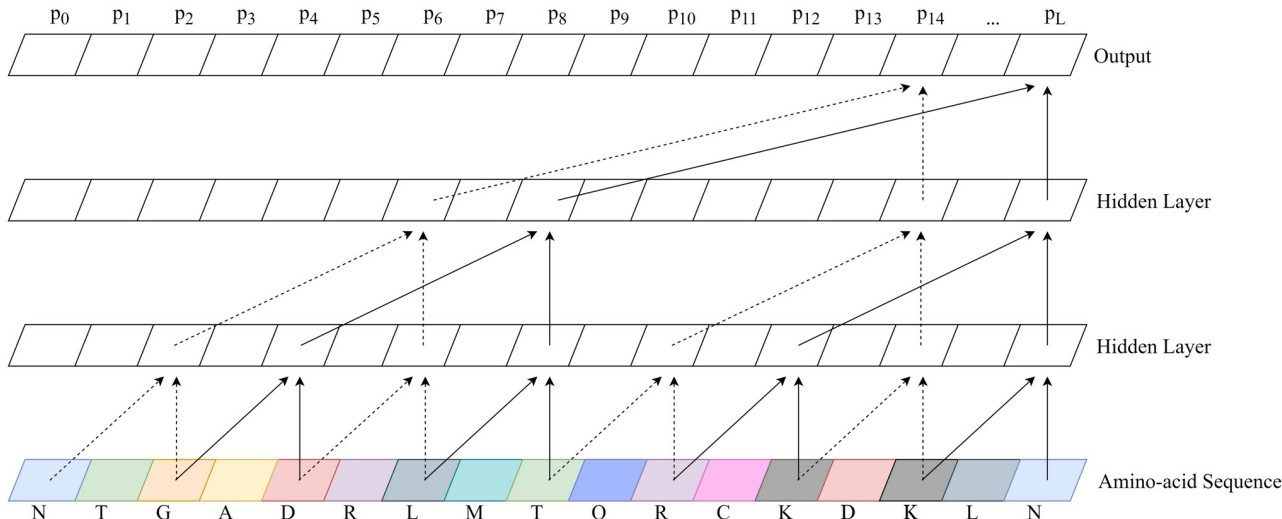

**Fig 1. Example temporal convolutional network.** A temporal convolutional network (TCN) with 2 hidden layers and a convolutional kernel size of 2. The number of connections exponentially increases as hidden layers are added, enabling a wide receptive field. Notice the output of a TCN is the same length as the input. Balrog's TCN used 8 hidden layers, a convolutional kernel size of 8, a dilation factor of 2, and 32 * L hidden units per layer where L is the length of the amino-acid sequence.

to improve gene scores based on the presence of conserved motifs in true proteins.

$$\text{Predicted gene probability} = \frac{1}{1 + e^{-x}}, \; x = \frac{1}{L}\sum_{i=1}^{L} ln\left(\frac{p_i}{1 - p_i}\right) \qquad (1)$$

Our gene model TCN used 8 hidden layers, 32 * L hidden units per layer, a dilation factor of 2, and a convolutional kernel size of 8. Dropout was performed on 5% of nodes during training to mitigate overfitting. We used adaptive moment estimation with decoupled weight decay regularization (AdamW) [17] to minimize loss during initial training, while final loss minimization was performed by stochastic gradient descent with a learning rate of $10^{-4}$ and Nesterov momentum of 0.90 [18, 19]. We performed all training on Google Colab servers with 32GB of RAM and a 16GB NVIDIA Tesla P100 GPU over the course of 48 hours.

## Training the translation initiation site model

Though not the main focus of this work, a good start site model provides a boost in accuracy for a prokaryotic gene finder. In bacteria, the initiation of translation is usually marked by a ribosome binding site (RBS), which manifests as a conserved 5-6 bp sequence just upstream of the start codon of a protein-coding gene. Experimentally-validated start sites are not available for the vast majority of bacterial genes, so we made the assumption (also used in previous methods [2]) that the annotated start sites of known genes would usually, but not always, be correct. Thus to create a RBS model, we extracted 16 nucleotides upstream and downstream from all annotated non-hypothetical gene start sites in the training set genomes. For each start site, we also found the closest downstream start codon within the gene and extracted the same sized windows for use as examples of false start sites.

Similar to the gene model, we trained a TCN on the positive and negative examples of gene start sites. A slightly smaller model was used due to the reduced complexity and length of the start site sequence data. Our start site model used 5 layers with 25 * L hidden units per layer

and a convolutional kernel size of 6. The model was trained for 12 hours on the same Google Colab server type as the gene model.

## Gene finding

A powerful gene sequence model is necessary for finding genes, but additional features such as open reading frame (ORF) length can also be taken into account. In particular, longer ORFs are more likely to be protein-coding genes, by the simple argument that a long stretch of DNA without stop codons is less likely, in random DNA sequence, than a short stretch. Balrog begins by identifying and translating all ORFs longer than a user-specified minimum. Its task is to determine for each of these ORFs whether it represents a protein-coding gene.

We also developed an optional kmer-based filter, using amino-acid sequences of length 10, which runs before the gene model to positively identify genes. This filtering procedure simply identifies all amino-acid 10-mers found in annotated non-hypothetical genes from the training data set and flags any ORF containing at least two of these 10-mers as a true protein. This initial step finds many common prokaryotic genes with a very high specificity and near-zero false positive rate.

Next, ORFs are scored by the pre-trained temporal convolutional network in the 3' to 5' direction. The region surrounding each potential start site of each ORF is then scored by the start site model. A directed acyclic graph is constructed for each contig, with nodes representing all possible ORFs. Edges are added between compatible ORFs overlapping by less than a user-specified minimum. To avoid creating a graph with $O(n^2)$ edges, we only connect a constant number of nodes to each node. Because prokaryotes are gene dense, we do not expect any large region with a significant number of non-gene ORFs. Therefore, we can keep the number of edges to $O(n * C)$ where C = 50 was empirically found to be sufficient for all tested genomes. Edge weights are calculated by a linear combination of the gene model score of the ORF, the gene start site model of the potential start site, a bonus for ATG vs. GTG vs. TTG start codon usage, and penalties for overlap depending on the 3'/5' orientation of the overlap.

The global maximum score of the directed acyclic graph is computed by finding the longest weighted path through the graph as shown in Fig 2. Because we are searching for the maximum score and some ORFs can receive negative scores, Dijkstra's algorithm does not work in this context [20]. Instead, we take advantage of the fact that our genome is implicitly topologically sorted to find the longest weighted path in two steps. First, we sweep forward along the genome, keeping track of the maximum attainable score at each node as well as its predecessor node. Then, we simply backtrack along the predecessors from the global maximum attainable score to find the longest weighted path. This is similar to finding the "critical path" in a task scheduling problem [21]. In practice, ORFs must only be connected locally to a relatively small set of other ORFs because no real prokaryotic genome should have a very large gap between genes. This makes the complexity of finding the maximum score scale linearly with the size of the genome. The highest scoring path through the graph represents the best predicted set of all compatible genes in the genome and is converted into an annotation file for the user.

To benchmark gene finding performance, Glimmer3 and Prodigal were run with default settings and allowed to train on each genome in the test set.

## Parameter optimization

In the spirit of building a data-driven model, nearly all parameters were optimized with respect to the data rather than being hand-tuned. Ten genomes were randomly selected from the training data set to use for optimization of weights used in the scoring function for genome graph construction.

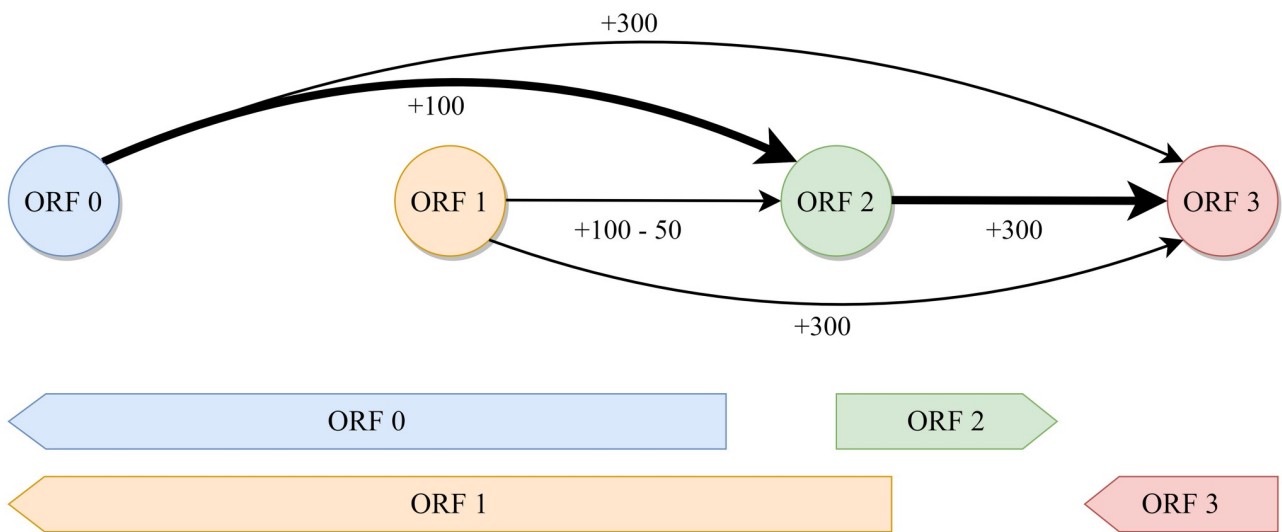

**Fig 2. Example ORF connection graph.** A directed acyclic graph with nodes representing open reading frames (ORFs) and edges representing possible connections. Each edge is weighted by the ORF score at the tip of the arrow minus any penalty for overlap. ORFs that overlap by too much are not connected. In this example, the maximum score is achieved by following the bolded path connecting 0-2-3. ORF 1 is not included because it is mutually exclusive with ORF 0 and results in a lower score due to overlap with ORF 2.

The score for each ORF node was calculated by a linear combination of features including the gene model score, start site model score, start site codon usage, and the length of the ORF. Additionally, final scores for edges between nodes are penalized by the length and direction of overlap, if any, between the connected ORFs. Depending on the type of overlap, per-base penalties are multiplied by the length of the overlap and subtracted from the edge connection score. Different penalties are learned for divergent overlap (3' to 3'), convergent overlap (5' to 5'), and unidirectional overlap (3' to 5' or 5' to 3').

This scoring system was used to combine features so the linear weights can be learned with respect to the data to maximize gene finding sensitivity. Optimization of all weights with respect to gene sensitivity was accomplished using a tree-structured Parzen estimator [22] and a covariance matrix adaptation evolution strategy [23]. Because ORFs do not need to be re-scored by the TCN during parameter optimization, only the graph construction and longest path finding steps must be iterated to maximize gene sensitivity. All optimization was carried out using the Optuna framework [24] over the course of 9 hours on two 10 core Intel Xeon E5-2680 v2 processors at 2.8GHz.

## Filtering with MMseqs2

Our gene model is tuned to maximize sensitivity to known genes without regard to the total number of predictions. In order to keep down the number of false positive predictions, users may optionally run a post-processing step with MMseqs2 [25]. In this step, we run all predictions against non-hypothetical protein coding gene sequence from a set of 177 diverse bacterial genomes. All reference genomes in this step do not share a genus with any of the test set organisms. Predictions are also run against the SWISS-PROT curated protein sequence database [26]. Any Balrog prediction that maps to a known gene with an E-value less than 0.001 is marked as a predicted gene. Finally, any gene below a set cutoff ORF score is discarded unless it was found by the kmer filter or MMseqs2. This process allows low-scoring predictions to be discarded as false positives while retaining many low-scoring genes that easily map to

conserved known genes. All genomes used in this step can be found in S3 Appendix. Fig 3 shows a flow chart with a broad overview of all steps performed by Balrog.

## Discussion

Balrog demonstrates that a data-driven approach to gene finding with minimal hand-tuned heuristics can match or outperform current state-of-the-art gene finders. By training a single gene model on nearly all available high-quality prokaryotic gene data, Balrog matches the sensitivity of widely used gene finders while predicting fewer genes overall. Balrog also requires no retraining or fine-tuning on any new genome.

Balrog predicted consistently fewer genes than both Prodigal and Glimmer3 on both the bacterial and archaeal genome test sets. The sensitivity of all three gene finders was nearly identical and likely well within the range of noise in our sample on average, though Prodigal appears to achieve higher sensitivity than both Balrog and Glimmer3 on high-GC% genomes. A stronger bias against short ORFs, similar to Prodigal's penalty on ORFs shorter than 250bp, may help Balrog perform better in genomes with particularly high GC content. However, incorporating a bias against small genes may provide higher specificity at the cost of sensitivity to small genes. Heuristics used by current gene finders, including default minimum ORF lengths of 90 for Prodigal and 110 for Glimmer3, have led to a blind spot around functionally important small prokaryotic proteins [27]. Balrog's default minimum ORF length is 60 nucleotides. Further work on finding small genes without significantly increasing false positive predictions may help illuminate this underappreciated category of prokaryotic genome function.

Our test set deliberately represented a near-worst-case scenario for Balrog, where no organism from the same genus was used to train the model. On organisms closely related to those in the large and diverse training set, we expect Balrog may perform better as a result of overfitting. Overfitting of a gene model in this context is a complex issue. Simply memorizing and aligning to all known genes can be thought of as the ultimate overfit model, yet that strategy would likely prove effective at finding conserved bacterial genes. Finding prokaryotic genes is not a standard machine learning task where memorization inevitably leads to higher generalization error. Conserved amino-acid sequences in prokaryotic genes may represent functionally important protein motifs and memorization of short amino-acid sequences as indicators of protein coding sequence may prove useful in finding genes even in novel organisms. Still, we attempted to be as fair as possible to competing gene finders by removing all organisms with a shared genus. We felt this should provide a conservative estimate of the true generalization error of our model to relatively distant genomes.

An alternative approach to training a universal protein model could use protein clusters to capture diversity in protein sequences with less redundancy than our whole-genome approach. However, we wanted our final evaluation metric to be as fair as possible to all gene finders and reflective of a real-world situation where a newly sequenced prokaryote would likely contain many proteins from many different clusters.

Balrog in its current form is relatively slow. While tools like Prodigal and GeneMarkS-2 may analyze a genome in a matter of seconds, Balrog may take minutes per genome. This is due to a wide range of factors including the complexity of the gene model and the optional gene filtering step with MMseqs2. Optimization of run time represents a possible future improvement for Balrog.

Balrog was designed primarily to find genes without much regard for identifying the exact location of their translation initiation site (TIS). TIS identification is a challenging problem with relatively little available ground-truth data. A reasonably accurate start site predictor

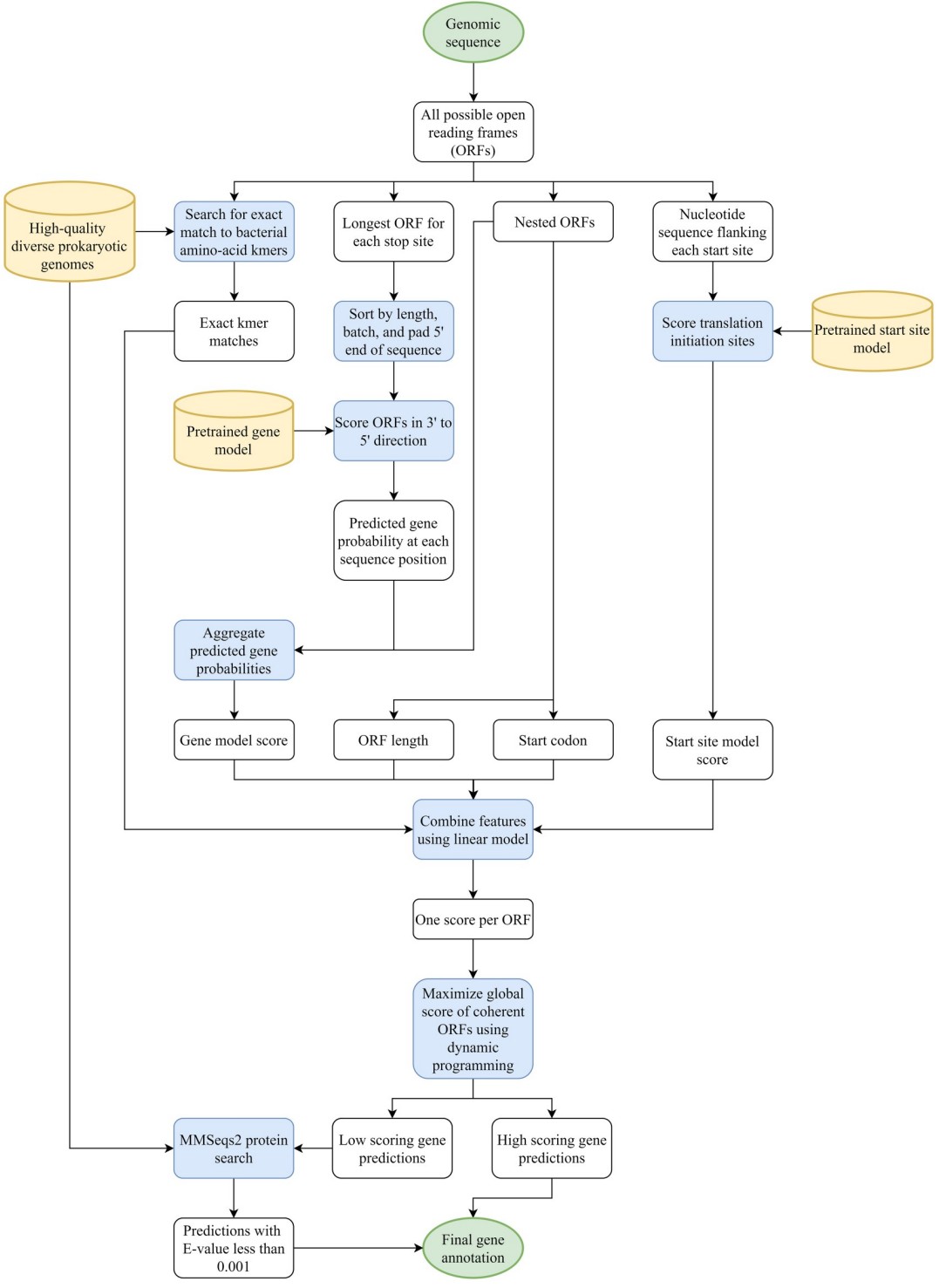

**Fig 3. Balrog gene finding flow chart.** A diagram showing all steps from genomic sequence in to gene predictions out. Green circles represent input and output data. White squares represent intermediate data. Blue squares represent processes. Yellow cylinders represent databases and pretrained models.

helps to guide a gene finder, so Balrog does include a small TIS model, but accurate start site prediction was not a primary focus of this work. Further complicating the issue, nearly all available start site locations are based solely on predictions of previous gene finders. Demonstrating true improvement in start site prediction would require comparing Balrog to other gene finders on a large ground-truth data set which is simply not currently available. Incorporating TIS models used by Prodigal or GeneMark may enable improvement in start site identification in the future.

## Supporting information

**S1 Appendix. Gene model testing organism information.** Full organism names and accession numbers of all genomes used in the gene finder comparison in Table 1.
(CSV)

**S2 Appendix. Gene model training organism information.** Full organism names and accession numbers of all genomes used to train the gene model.
(CSV)

**S3 Appendix. MMseqs2 and kmer filter organism information.** Full organism names and accession numbers of all genomes used in the protein kmer and MMseqs2 filtering steps.
(CSV)

## Acknowledgments

We would like to thank Christopher Pockrandt for helping distribute the C++ version of Balrog, Jennifer Lu and Alaina Shumate for helping brainstorm cool program names, everyone on the Center for Computational Biology Slack channel for voting on said cool names, Martin Steinegger for helpful conversations and creating MMseqs2, @genexa_ch for providing via Twitter a small set of diverse GTDB genomes on which the kmer filter and MMseqs2 are run, and everyone in the S. Salzberg and M. Pertea labs.

## Author Contributions

**Conceptualization:** Markus J. Sommer, Steven L. Salzberg.

**Data curation:** Markus J. Sommer.

**Formal analysis:** Markus J. Sommer.

**Funding acquisition:** Steven L. Salzberg.

**Investigation:** Markus J. Sommer.

**Methodology:** Markus J. Sommer, Steven L. Salzberg.

**Project administration:** Steven L. Salzberg.

**Software:** Markus J. Sommer.

**Supervision:** Steven L. Salzberg.

**Validation:** Markus J. Sommer.

**Writing – original draft:** Markus J. Sommer.

**Writing – review & editing:** Markus J. Sommer, Steven L. Salzberg.

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
