## [Decision Letter · Decision Letter 0]

21 Oct 2020

Dear Mr. Sommer,

Thank you very much for submitting your manuscript "Balrog: A universal protein model for prokaryotic gene prediction" for consideration at PLOS Computational Biology.

As with all papers reviewed by the journal, your manuscript was reviewed by members of the editorial board and by several independent reviewers. In light of the reviews (below this email), we would like to invite the resubmission of a significantly-revised version that takes into account the reviewers' comments.

We cannot make any decision about publication until we have seen the revised manuscript and your response to the reviewers' comments. Your revised manuscript is also likely to be sent to reviewers for further evaluation.

Sincerely,

Christos A. Ouzounis

Associate Editor

PLOS Computational Biology

William Noble

Deputy Editor

PLOS Computational Biology

Reviewer's Responses to Questions

**Comments to the Authors:**

Reviewer #1: I am glad to see some new tool developed using deep learning to predict proteins from prokaryote genomes and this sounds a nice tool to improve the accuracy and easily used without training for specific taxonomic units like other tools, e.g. prodigal and prokka. I would like to test it by myself, but failed in using the web server provided by the authors, even failed in uploading my sequences. I suppose the model file is large hindering distribution of the tool. Standalone version is much helpful.

The writing is good, then I still have other concerns.

1) As we know, prokaryote genome sequnces are largely biased in sequncing for some pathogens. Then the data set for taining is not balanced.

2) for prokaryote genomes, the difference of gene numbers within the same species, that is, different populations/strains, is large because of HGT or other reasons resulting in quite difference if accessory genomes. Why the authors select proteins for training based on the rule of picking up genomes and then determine the proteins. It looks like that the authors need to select all prokaryote genomes with high qualtiy and then cuurate pangenome to cluster these proteins for traing your model. Another option is that the author could extract high quality of protein seuqnces of porkaryotes from known databases, e.g. uniprot.

3）I am not sure the rule to make non-hypothetical genes only based on a description containing “hypothetical” or “putative”. This is really coarse.

4) two figures are too simple to express clearly what was done by authors.

Reviewer #2: The authors developed a method for gene prediction in prokaryotes, Balrog, which is based on deep convolutional neural networks (CNNs) and was trained on 3290 genomes and tested on 36. To focus the test results on non-trivial cases, no genomes from the same genus as any of the test were allowed in the training set. The method employs recent technological developments of using CNNs in sequence modeling (Bai, Kolter, Koltun, 2018). First, a CNN is trained to predict for every position of a translated amino acid sequence whether translation is in the right frame or not. This is the heart of the method. Second, a CNN for predicting translation initiation sites is trained on the 32 nucleotide long sequences around each start sites of the non-hypothetical proteins in the training set. Third, to avoid making contradicting ORF calls (e.g. strongly overlapping ones), the longest weighted path through a directed acyclic graph is computed, in which nodes represent possible ORFs and nodes are connected by edges if the ORFs do not overlap too much.

Balrog achieves very similar sensitivity as the gold standard tools Prodigal and Glimmer3, and it has 11% and 30% fewer likely false predictions than Prodigal and Glimmer3, respectively. Balrog takes 5-10 minutes to process a typical bacterial genome on a GPU, whereas Prodigal takes a few seconds at most on a single CPU core.

The results are a bit disappointing considering the big advances that deep learning has afforded in many bioinformatic applications. However, the study is interesting for two reasons. First, if the slight improvements hold true with an unbiased benchmark, they would be a worthwhile improvement of prediction accuracy. Second, the study demonstrates how to use state-of-the-art deep learning methods for the task of gene prediction.

Major points:

1) It is unclear to what degree the training set is biased by the fact that many gene annotations in the training genomes are also produced by bioinformatic prediction tools. Since Glimmer3 and Prodigal have been the standard tools for gene prediction since 1998 and 2010, respectively, it is likely that most of the 'extra" genes annotated as hypothetical were actually predicted found by Glimmer3 or Prodigal. It is therefore not surprising at all that Glimmer3 and Prodigal would find more such 'extra' genes than a tool such as Balrog using a very different methodology.

The authors need to construct a benchmark that can correct for such biases or at least estimate them. One option could be to test on genomes that have been annotated using experimental data such as RNA-seq, CAGE-seq or the like.

2) It would be important to get more information on how much this very highly parameterized method can generalize beyond the genus. The benchmark should therefore be repeated with training sequences from which all genomes from the same family / order of any of the test genomes have been excluded.

Minor points:

3) The Methods do not mention what dilation sizes were used in the gene model CNN. d = 2^i ?

4) To train the start site model, negative training examples were taken to be the start site codons after the annotated start site of the positive training ORFs. Isn't that quite risky since start sites are notoriously hard to annotate and might be frequently wrong? Wouldn't it be better to use start codons within the negative ORF training examples?

5) Whereas it is stressed in the abstract that Prodigal and Glimmer3 need to be pretrained on each genome to achieve optimal results, the Methods section does not mention if such pretraining was employed.

6) Please explain why 'Efficiently training a temporal convolutional network requires sequences of the same 96 length.'

7) Why is Balrog so slow? I count 20 * 8 * 32 * 8 = 41920 parameters. Since predictions can be done in parallel on the GPU, that should take a few seconds, not minutes, for the few ten thousand translated ORFs longer than 60 codons. Could it be that each convolutional filter is not only computed once per input window of length k, as it should, but 100-k+1 times (where 100 is the length of the sequences used for training)?

8) Line 13: Delete '32 ∗ L hidden units per layer, and'

9) Please comment in the discussion on why you did not use a transformer architecture.

**Have all data underlying the figures and results presented in the manuscript been provided?**

Reviewer #1: Yes

Reviewer #2: Yes

PLOS authors have the option to publish the peer review history of their article (what does this mean?). If published, this will include your full peer review and any attached files.

Reviewer #1: No

Reviewer #2: No
---

## [Decision Letter · Decision Letter 1]

19 Jan 2021

Dear Mr. Sommer,

We are pleased to inform you that your manuscript 'Balrog: A universal protein model for prokaryotic gene prediction' has been provisionally accepted for publication in PLOS Computational Biology.

Best regards,

Christos A. Ouzounis

Associate Editor

PLOS Computational Biology

William Noble

Deputy Editor

PLOS Computational Biology

Reviewer's Responses to Questions

**Comments to the Authors:**

Reviewer #1: No further comment.

Reviewer #2: The authors have addressed all reviewer comments satisfactorily. I particularly appreciate providing open-source C++ code that can run on CPUs.

**Have all data underlying the figures and results presented in the manuscript been provided?**

Reviewer #1: Yes

Reviewer #2: Yes

PLOS authors have the option to publish the peer review history of their article (what does this mean?). If published, this will include your full peer review and any attached files.

Reviewer #1: No

Reviewer #2: No

---

## [Editor Report · Acceptance letter]

15 Feb 2021

PCOMPBIOL-D-20-01618R1 

Balrog: A universal protein model for prokaryotic gene prediction

Dear Dr Sommer,

I am pleased to inform you that your manuscript has been formally accepted for publication in PLOS Computational Biology. Your manuscript is now with our production department and you will be notified of the publication date in due course.

With kind regards,

Alice Ellingham
